# Role of Ferroptosis in Regulating the Epithelial–Mesenchymal Transition in Pulmonary Fibrosis

**DOI:** 10.3390/biomedicines11010163

**Published:** 2023-01-09

**Authors:** Hong Ling, Hong Xiao, Ting Luo, Huicai Lin, Jiang Deng

**Affiliations:** 1Key Laboratory of Basic Pharmacology of Ministry of Education and Joint International Research Laboratory of Ethnomedicine of Ministry of Education, Zunyi Medical University, Zunyi 563006, China; 2Key Laboratory of Basic Pharmacology of Guizhou Province, Zunyi Medical University, Zunyi 563006, China; 3Department of Pharmacology, School of Pharmacy, Zunyi Medical University, Zunyi 563006, China

**Keywords:** pulmonary fibrosis, ferroptosis, epithelial-mesenchymal transition transforming growth factor beta/Smad signaling pathway, nuclear factor erythroid 2–related factor 2 signaling pathway, Wnt signaling pathway

## Abstract

Idiopathic pulmonary fibrosis is a chronic interstitial lung disease whose pathogenesis involves a complex interaction of cell types and signaling pathways. Lung epithelial cells responding to repeated injury experience persistent inflammation and sustained epithelial–mesenchymal transition (EMT). The persistence of EMT-induced signals generates extracellular matrix accumulation, thereby causing fibrosis. Ferroptosis is a newly characterized iron-dependent non-apoptotic regulated cell death. Increased iron accumulation can increase iron-induced oxidant damage in alveolar epithelial cells. Studies have demonstrated that iron steady states and oxidation steady states play an important role in the iron death regulation of EMT. This review summarizes the role of ferroptosis in regulating EMT in pulmonary fibrosis, aiming to provide a new idea for the prevention and treatment of this disease.

## 1. Introduction

Pulmonary fibrosis is a progressive, irreversible and usually fatal diffuse lung interstitial disease. Currently, there is no effective radical cure and the prognosis is extremely poor; the average life expectancy is 3–5 years after diagnosis [1]. The main pathological features include alveolar structure destruction, lung fibroblast proliferation and extracellular matrix (ECM) deposition, which result in decreased lung compliance and gas exchange disorders, ultimately leading to extensive scarring, lung failure and even death [2,3,4]. The inflammatory cells release some key regulatory factors, such as pro-fibrotic cytokines, chemokines, and growth factors, which can induce the epithelial–mesenchymal transition (EMT) [5,6]. Importantly, EMT contributes to the early development of interstitial fibrosis through paracrine signaling from the alveolar epithelium to potential fibroblasts [7]. The initiation and progression of pulmonary fibrosis involve many factors, which may be related to long-term smoking, viral infection, genetics, aging and environmental factors [8]. The impact of the coronavirus disease 2019 (COVID-19) infection on lung diseases is noteworthy in recent years. Severe acute respiratory syndrome coronavirus 2 (SARS-CoV-2) is a highly transmissible and pathogenic coronavirus that emerged in late 2019 and has caused a pandemic of acute respiratory disease named ‘coronavirus disease 2019’ (COVID-19) which threatens human health and public safety. SARS-CoV-2 enters the nasal epithelium, spreads to the respiratory system and causes diffuse alveolar damage by binding to the angiotensin converting enzyme-2 (ACE-2) receptors on the surface of lung epithelial cells [9]. SARS-CoV-2 can lead to acute lung inflammation by mobilizing iron in the vascular space to activate the hepcidin-Fpn pathway and promote ferroptosis [10]. Pulmonary fibrosis is further promoted by alveolar thrombosis and airway inflammatory viral injury, and SARS-CoV-2 can also induce pulmonary fibrosis by promoting the upregulation of TGF-β and other pro-fibrosis signaling molecules [11].

Ferroptosis is an iron-dependent non-apoptotic regulated cell death [12,13]. The accumulation of excessive intracellular iron, depletion of glutathione (GSH), inactivation of glutathione peroxidase 4 (GPX4) and upregulation of lipid peroxidation are essential to the development and progression of ferroptosis [14]. Followingacute inflammation and injury, the lungs undergo repair and remodeling to restore homeostasis, accompanied by fibrosis and scar formation which may lead to pulmonary fibrosis [15]. In the pathological process of acute lung injury, the release of various reactive oxygen species and the generation of free radicals can damage alveolar epithelial cells. Iron overload can further promote the transformation of hydrogen peroxide into free radicals through the Fenton reaction, thus increasing cytotoxicity and promoting the occurrence and development of acute lung injury [16]. Meanwhile, it has been reported that there exist eight ferroptosis-related genes signatures (including NRAS, EMP1, SLC40A1, MYC, ANGPTL4, PRKCA, MUC1, and GABARAPL1) in the bronchoalveolar lavage fluid of patients with idiopathic pulmonary fibrosis (IPF) that are associated the diagnosis and prognosis of IPF [17]. He J et al. [18] found 1692 differential genes, of which 20 genes were associated with ferroptosis, by comparing IPF lung tissue with the normal lung tissue of mice. These genes were divided into driving factors of ferroptosis (CA9, EPAS1, CDO1, CDKN2A, and ALOX15), inhibiting factors of ferroptosis (TP63, CAV1, PROM2, and JUN) and markers of ferroptosis (NOS2, HNF4A, RGS4, SLC2A1, GDF15, SLC2A12, NGB, DRD5, and GPX2). Meanwhile, increased iron levels directly promoted the proliferation, proinflammatory cytokines and ECM response of lung fibroblasts. However, deferoxamine reduces the number of Tfr1+ macrophages with M2-like phenotypes in pulmonary fibrosis induced by bleomycin. The M2-like macrophages are known to play an important role in fibrosis [19]. Studies have shown that iron chelators ameliorate pulmonary fibrosis induced by bleomycin and alleviate leukocyte migration in mice. Yuan L. et al. [20] found that dihydroquercetin plays an anti-fibrosis role by inhibiting iron death in human bronchial epithelial cells. Pei Z. et al. [21] showed that bleomycin and lipopolysaccharide directly induce iron overload and ferroptosis in lung epithelial cells in the early inflammatory period. Transferrin receptor protein 1 promotes the transition from fibroblasts to myofibroblasts in advanced fibrosis through moderate intracellular accumulation of unstable ferrous iron mediated by the TGF-β-TAZ-TEAD signaling pathway. This suggests that ferroptosis may be involved in the development and progression of IPF.

The question is whether ferroptosis promotes pulmonary fibrosis by regulating EMT. Increased iron accumulation makes alveolar epithelial cells vulnerable to iron-induced oxidative damage [22]. An imbalance between oxidation and antioxidant has been observed in patients with pulmonary fibrosis, and iron deposition has been observed in lung tissue sections [23]. Meanwhile, iron metabolism may regulate transforming growth factor-β (TGF-β)-induced EMT through reactive oxygen species (ROS) production in the alveolar epithelium, after which pulmonary fibrosis occurs [24]. Furthermore, in vivo TGF-β1 induces ROS production in epithelial cells and inhibits antioxidant enzymes, leading to redox imbalance. ROS, in turn, induces or activates TGF-β1 and causes pulmonary fibrosis, creating a vicious cycle [25]. Understanding the role of ferroptosis in pulmonary fibrosis may provide a novel therapeutic direction for the prevention and treatment of this disease.

## 2. EMT and Pulmonary Fibrosis

The development of pulmonary fibrosis results from a complex interplay between epithelial cells, fibroblasts, immune cells and endothelial cells. Recently, there is increasing evidence that the alveolar epithelium plays a central role [26]. Senescent cells are the driver of IPF, and their dysfunction plays a key role in the activation and type-Ⅱ alveolar epithelial proliferation of lung fibroblasts [27]. TGF-β is upregulated and activated in fibrotic diseases, inducing myofibroblast transdifferentiation and promoting EMT of type-Ⅱ alveolar epithelial cells [28].

EMT is a process by which epithelial cells lose their cell–cell adhesion and apico-basal polarity and acquire mesenchymal properties that migrate, invade and produce ECM components [29,30]. The common markers of EMT are the loss of E-cadherin and cytokeratin, as well as the upregulation of matrix metalloproteinase, vimentin and α-smooth muscle actin (α-SMA). EMT plays a central role in organ fibrosis and cancer progression because it involves numerous morphological features of hyperproliferative diseases, such as cell plasticity, anti-apoptosis, dedifferentiation and proliferation [31,32]. The occurrence of pulmonary fibrosis is related to the upstream-related signaling pathways of EMT, and the occurrence and development direction of pulmonary fibrosis can be inhibited by intervening or blocking the relevant effector molecules of EMT [33,34]. EMT is divided into three types according to the specific biological environment. Type I is primary and occurs in the early stages of zygote implantation, embryogenesis and organogenesis. Type II is secondary and occurs during trauma healing, tissue regeneration and organ development. Type III occurs during tumor metastasis, and malignant epithelial cells acquire a migration phenotype associated with tumor invasion and metastasis. Pulmonary fibrosis is classified as a type II EMT [7,35,36].

## 3. Ferroptosis and EMT

### 3.1. Discovery of Ferroptosis

Ferroptosis is a type of programmed non-apoptotic cell death that is mainly caused by the imbalance between oxidative stress and antioxidant response and is characterized by the accumulation of iron-dependent lipid peroxides [37,38]. A variety of organelles—including mitochondria, the endoplasmic reticulum, the Golgi apparatus and lysosomes—are involved in the regulation of ferroptosis which manifests as reduced mitochondrial volume, increased mitochondrial membrane density, decreased or disappeared mitochondrial crest, increased ROS in the cytoplasm and mitochondrial outer membrane rupture [39]. It is caused by increased ROS levels due to increased intracellular iron concentration and lipid peroxidation due to depletion of the antioxidant GSH [40]. Ferroptosis is closely related to respiratory diseases, cancer, nervous system diseases and cardiovascular diseases, among others. In recent years, studying the mechanism of ferroptosis has become a new direction for the treatment of many diseases [41,42,43,44]. GSH biosynthesis and the normal function of phospholipid hydroperoxide glutathione GPX4 are key to the control of ferroptosis [45]. As a fourth member of the selenium-containing GPX family, GPX4 suppresses lipid peroxidation and oxidative stress-related cell death, which occurs when GPX4 is reduced or inactivated and is followed by mitochondrial damage [46]. As the most upstream component of the xc-/GSH/GPX4 axis, the transmembrane cystine-glutamate reverse transport system xc- is the heterodimer amino acid transporter family member composed of light chain xCT (SLC7A11) and heavy chain 4F2 (SLC3A2) [47]. The primary role of ferroptosis is the production of GPX4 catalyzed by active iron, which is counteracted by endogenous levels of system xc-. The oxidized form of cysteine exchange glutamate at a ratio of 1:1 and extracellular glutamate are competitive inhibitors of cysteine uptake.

### 3.2. Role of Ferroptosis in EMT

Some ferroptosis inducers such as erastin, sorafenib, sulfasalazine and glutamate can drain GSH and inactivate the enzymatic activity of GPX4 by blocking the import of cystine by the system xc- [48]. Upregulation of ChaC glutathione-specific γ-glutamyl cyclotransferase 1 (CHAC1) gene expression provides a selective pharmacodynamic marker for systemic xc- inhibitor-induced ferroptosis (CHAC1/BOTCH) with γ-glutathione aminoacyl cyclotransferase activity, and reduces intracellular GSH levels by digesting glutathione into 5-oxoproline and cysteine glycine dipeptides [49]. GPX4, as an antioxidant protein with glutathione peroxidase activity, is mainly responsible for phospholipid oxidation and ROS production during ferritization [50,51]. Excess free iron content reduces GPX4 activity, which in turn leads to GSH depletion. Decreased GPX4 activity is associated with increased ROS. When GSH is depleted, the ability of cells to remove ROS decreases, leading to membrane oxidation and ultimately to ferroptosis [52]. The expression of GPX4 was decreased in bleomycin-induced pulmonary fibrosis in mice, and the fibrosis was more severe when GPX4 gene was knocked out. Therefore, GPX4 regulation of pulmonary fibrosis induced by bleomycin in mice can be attributed to changes in ferroptosis [53]. ROS regulates the AKT/mammalian target of rapamycin (mTOR) signaling, and mTOR plays a role in TGF-β1-induced EMT [54]. Mefunidone, an antifibrotic drug, can reduce ROS production and inhibits the TGF-β1/Smad pathway, phosphorylation levels of ERK1/2, JNK and P38, thereby inhibiting EMT [55]. Studies have reported that NaHS can prevent cigarette smoke extract-induced oxidative damage in bronchial epithelial cells, which is mediated by decreased ROS production and increased antioxidant enzyme activity [56]. In addition, higher levels of ROS trigger DNA damage, p53 activation, cell cycle blockade and cell death due to apoptosis and/or necrosis, all of which may be important in the ultimate fibrotic response [57].

During ferroptosis, ROS produced during oxidative stress can induce autophagy. The degradation of ferritin by autophagy plays an important role in the pathogenesis of ferroptosis. At the same time, autophagy can regulate the progression of EMT [58]. ROS plays a dual role in EMT; moderate ROS promotes EMT, and large ROS production reverses EMT [59]. Erastin stimulates iron overload in mouse lung epithelial cell lines in vitro which causes an oxidative stress response in the cytoplasm and increases cell death, activates autophagy and secretes pro-fibrotic factors, resulting in EMT [22]. However, ferritinophagy-mediated ROS production contributing to the EMT inhibition and ferritin phagocytosis refers to the selective autophagic degradation of ferritin, which leads to the accumulation of cytosolic iron in the form of ferrous iron, and ultimately to the occurrence of ferroptosis [60,61]. Ferritinophagic flux (nuclear receptor coactivator 4 [NCOA4]/ ferritin) can regulate EMT; that is, NCOA4 is involved in the EMT process. EMT inhibition induced by 2,2′-di-pyridylketone hydrazone dithiocarbamate butyric acid (DpdtbA) is involved in the production of ROS and activation of prolyl hydroxylase domain protein 2 (PHD2) mediated by ferritin phagocytosis, indicating that ferritin phagocytosis-mediated accumulation of ferrous ions leads to the activation of PHD2 and p53 and EMT inhibition [62]. This raises the following question: Do ferroptosis and EMT occur simultaneously? Ferrostatin-1, a ferroptosis inhibitor, can cause the loss of GPX4 function, inducing ferroptosis in mesenchymal-state GPX4-knockout cells, whereas GPX4-knockout cells in the epithelial state remain unaffected, resulting in mesenchymal cells that are more likely to cause ferroptosis [63]. Therefore, ferroptosis and EMT do not occur simultaneously.

Yao et al. [64] found that some ferroptosis markers (GPX4, SCP2, and CAV1) have strong regulatory effects on EMT. SCP2 is a driver of ferroptosis, promoting other ferroptosis drivers (PRKAA1, PRKAA2) and EMT markers (N-cadherin) but inhibiting ferroptosis inhibitors such as GPX4 and caveolin-1 (CAV1). CAV1 is an integral membrane protein involved in cell signal transduction and transport. It is highly expressed in adipocytes, endothelial cells, fibroblasts and cancer cells [65]. Knockdown of CAV1 can reduce the expression of GPX4, and it is negatively correlated with ROS. CAV1 is a negative regulator of TGF-β1 activity; it reduces the subsequent effects of TGF-β1, leading to a decrease in collagen type I and fibronectin and an increase in MMP mRNA expression [66]. CAV1-deficient peritoneal mesothelial cells acquire the ability to degrade ECM, which is characteristic of cells that have undergone EMT. In the absence of CAV1, expression of E-cadherin and other proteins located at cell junctions is reduced, so CAV1 depletion promotes the EMT phenotype [67]. Meanwhile, the cross-talk between DPP-4, CAV1 and integrin β1 plays a key role in DPP-4 and TGF-β1-induced signal transduction and EMT induction in epithelial cells [68].

TGF-β1 may directly inhibit the cystine/glutamate anti-transporter system xc- and promote ferroptosis [69]. Short-term treatment of fibroblasts with TNF induces ROS and promotes ferroptosis. However, long-term exposure to TNF failed to further induce ROS, inhibit NAPDH oxidase and promote cystine uptake and GSH biosynthesis to protect fibroblasts from ferroptosis [70]. Meanwhile, TNF-α enhanced TGF-β-induced EMT in A549 cells. A549 cells treated with TGF-β and TNF-α exhibit a more severe loss of cell adhesion characteristics and obtain the migration and invasion abilities unique to mesenchymal cells [71]. TNF-α is usually produced more in the early stages of inflammation, and high levels of TNF-α can increase collagen deposition and lead to alveolar damage. Therefore, TNF-α is related to the early stage of pulmonary fibrosis and EMT and ferroptosis can be regulated by TNF [72].

## 4. Multiple Signaling Pathways Associated with Ferroptosis Regulate EMT in Pulmonary Fibrosis

The differential gene expression between epithelioid cells and fibroblasts has revealed that TGF-β and Wnt signaling pathways are most affected, and that both pathways are critical for the EMT and mesenchymal–epithelial transition processes [73]. Other signaling pathways, such as the Notch, phosphoinositide 3-kinase-AKT-mTOR, Janus kinase 2-signal transducer and activator of transcription 3, epidermal growth factor receptor-Ras-mitogen-activated protein kinases and Hippo signaling pathways can also induce or control the EMT process and participate in the process of pulmonary fibrosis [74,75]. Among them, the TGF-β/Smad, nuclear factor erythroid 2–related factor 2 (Nrf2) and Wnt signaling pathways are also involved in ferroptosis [21,76,77]. The regulation of ferroptosis in EMT-induced pulmonary fibrosis involves the TGF and Nrf2 signaling pathways, while the Wnt pathway is worth exploring.

### 4.1. TGF-β/Smad Signaling Pathway

The TGF-β signaling pathway plays a central role in the pathogenesis of idiopathic pulmonary fibrosis. TGF-β binds to the latency-associated peptide (LAP) to form the small latent complex (SLC) by non-covalent bonds and inhibits TGF-β binding to its receptor [78,79]. Then, the complex binds to the latent TGF-β binding protein in the endoplasmic reticulum to form the large latent complex (LLC) [80]. αvβ6 causes increased TGF-β autocrine signaling during EMT by activating TGF-β1 [81,82]. The binding of TGF-β to TGFβR-II causes phosphorylation of TGFβR-II, which activates TGFβR-I and then phosphorylates Smad2 and Smad3 in the cells. Phosphorylated Smad2 and Smad3 interact with Smad4 and translocate into the nucleus to regulate target gene transcription by binding to DNA-binding transcription factors [81,83]. TGF-β1 has a particularly prominent role in inducing the differentiation of progenitors into myofibroblasts that rapidly produce large amounts of ECM to maintain the repair of damaged tissues [84].

Zinc finger E-box-binding homeobox 1 (ZEB1), Ras-selective lethal 3 (RSL3), SETDB1, long non-coding RNA (lncRNA), erastin and N-Myc downstream regulated 1 (NDRG1) can regulate ferroptosis and induce EMT through the TGF-β/Smad signaling pathway. ZEB1 is one of the key factors controlling EMT [85]. TGF-β promotes EMT in cancer cells by inducing ZEB1 while increasing susceptibility to GPX4 inhibitors and statins [86]. In addition, ZEB1 directly inhibits GPX4 promoter transcription activity and GPX4 activity by binding to the E-box motif, and promotes ROS accumulation in vitro [87]. Notably, ROS accumulation promotes the increase of intracellular unstable iron and thus induces the occurrence of ferroptosis [88]. As a ferroptosis activator, RSL3 can cause the occurrence of cellular ferroptosis and idiopathic pulmonary fibrosis by directly inhibiting GPX4, which is associated with the redox imbalance in the lung. As such, the expression of GPX4 is decreased in lung tissue of idiopathic pulmonary fibrosis. When GPX4 is downregulated, TGF-β can induce fibroblast differentiation in vitro and generate pulmonary fibrosis [89].

SETDB1 is a specific methyltransferase for histone H3K9, which directly regulates the expression of Snail1 and indirectly regulates the expression of E-cadherin through the modification of H3K9me3. Overexpression of SETDB1 promotes a TGF-β-induced increase in iron levels and further downregulates GPX4, promoting the occurrence of ferroptosis [25].

Knockdown of lncRNA H19 attenuates pulmonary fibrosis in vitro and in vivo by regulating the microRNA 140-TGF-β/Smad3 signaling pathway. Meanwhile, lncRNA ZEB1-AS1 promotes pulmonary fibrosis by enhancing ZEB1-mediated EMT, and lncRNALOC344887 is a novel anti-fibrosis Nrf2 target gene [90].

NDRG1 is a gene related to iron chelator regulation, and knockdown of NDRG1 leads to the upregulation of GPX4 and xCT, which can induce the occurrence of ferroptosis [59]. Overexpression of NDRG1 decreases the expression of Smad2 and phosphorylated Smad3, thus inhibiting the TGF-β/Smad pathway. The Smad complex binds to the snail and slug promoters to promote the increased expression of E-cadherin and the formation of the adhesion complexes, which inhibits cell–cell adhesion and cell migration, ultimately inhibiting pulmonary fibrosis [91].

Li M et al. [92] compared the bronchoalveolar lavage fluid of IPF patients with that of normal subjects and found that differentially expressed genes (DEGs) associated with risk scores are mainly enriched in epithelial cell proliferation and ECM tissue. In addition, these DEGs have been identified in the most critical pathways for the development of idiopathic pulmonary fibrosis, such as cytokine-cytokine receptor interaction, TGF-β signaling, focal adhesion and the ECM-receptor interaction signaling pathways. Among them, ACO1 is associated with pulmonary fibrosis, and cytoplasmic aconitase 1/ iron regulatory protein 1 (ACO1/IRP1), a bifunctional protein expressed in the cytoplasm, played a role in regulating iron homeostasis in cells [93,94]. Previous studies found that the CAV1 gene was highly correlated with ferroptosis through protein–protein interaction analysis; CAV1 is widely found in lung tissues, including alveolar epithelial cells, endothelial cells, fibroblasts and smooth muscle cells [18]. CAV1 inhibits collagen formation in lung fibroblasts and attenuates Smad3 nuclear translocation, which suggests that the TGF-β/Smad pathway is closely related to CAV1 [95].

Cell death is a critical issue in radiation-induced pulmonary fibrosis (RILF). Radiation not only damages cellular DNA but also induces ROS production, which causes inflammation and fibrosis in lung tissue. GPX4 is mainly expressed in airway epithelial cells. Furthermore, radiation induces ferroptosis in airway epithelial cells and upregulates the release of inflammatory cytokines including TGF-β1, resulting in collagen deposition in lung parenchyma and promoting lung fibrosis [96]. Liproxstatin-1, as an inhibitor of ferroptosis, improves RILF by reducing lung collagen deposition and urinary hydroxyproline content and significantly downregulating ROS and TGF-β1 levels [97] (Figure 1). 

### 4.2. Nrf2 Signaling Pathway

Nrf2 is an important transcription factor that resists oxidative stress and exerts anti-inflammatory responses by coordinating the transcription of the target genes of the antioxidant response element (ARE) [90,98]. Usually, as an inhibitor of Nrf2, Kelch-1ike ECH-associated protein l (Keap1) binds to Nrf2 and degrades Nrf2 through ubiquitination. Upon exposure to oxidative stress, ROS inhibits Keap1 activity by inducing conformational changes in Keap1, which in turn hinders the ubiquitination of Nrf2 by cullin 3 [99,100]. Nrf2 dissociates from Keap1 and translocates into the nucleus to form a heterodimer with small Maf or Jun proteins which binds to the ARE to transcriptionally activate its target genes and restore cellular redox homeostasis [90,101,102,103].

Nrf2 plays a key role in maintaining the balance of GSH in the mitochondria. The Keap1/Nrf2 signaling pathway can mediate intracellular oxidative stress and regulate genes associated with the oxygen-scavenging free radical [104]. Nrf2 can enhance the expression of GSH biosynthase and reductase, which in turn inhibits mitochondria-derived ROS [103]. Knockdown of Nrf2 significantly reduces the protein expression levels of SLC7A11 and heme oxygenase 1 (HO-1), promotes the accumulation of lipid peroxidation and causes ferroptosis. In the presence of reduced iron, ferrostatin-1 eliminates lipid hydrogen peroxide and produces the same anti-ferroptosis effect as GPX4 to protect cells [105]. Ferrostatin-1 ameliorates lung injury by improving pulmonary edema, inhibiting lipid peroxidation and increasing the viability of epithelial cells [106]. The expression of Nrf2, HO-1, NAD(P)H quinone dehydrogenase 1 and epoxide hydrolase is significantly reduced, and the downregulation of Nrf2 is associated with the upregulation of α-SMA and collagen in idiopathic pulmonary fibrosis patients [99].

xCT knockdown results in the downregulation of Nrf2 and Keap1, whereas xCT overexpression has no effect on Nrf2 and Keap1 mRNA levels, so xCT is an effective downstream target of Nrf2 [107]. Atorvastatin inhibits Nrf2, thereby inhibiting the expression of system xc- and GPX4, resulting in severe damage to the antioxidant system and ferroptosis [108]. Inhibition of system xc-, depletion of GSH and enhancement of oxidative stress promote ferroptosis and EMT [89]. Therefore, EMT can be inhibited by regulating the system xc-, and pulmonary fibrosis can be suppressed.

The transcription factor BTB and CNC homology 1 (Bach1) bind to heme and participate in the regulation of oxidative stress response and metabolic pathways related to heme and iron [109]. Bach1 can regulate EMT by affecting intercellular adhesion genes such as claudin 3 (CLDN3) and CLDN4. Other transcription factors such as forkhead box A1 (FOXA1) and Snail2 can also directly affect the EMT process by forming a gene regulatory network (GRN) composed of transcription factors and their target genes [110]. Knockdown of Bach1 can reduce the gene expression of E-cadherin and promote the occurrence of EMT, and FOXA1 mediates the connection between Bach1 and E-cadherin [111]. Meanwhile, Bach1 directly activates Snail2, which encodes a prototypic transcription factor to initiate EMT by inhibiting cell–cell adhesion and promoting stem cell function [112]. Bach1 inhibits Nrf2 signaling by binding to competitive dimers and EpRE sites in the target gene promoters with Nrf2. Knockout of Bach1 enhances the expression of Nrf2 regulatory genes, especially HO-1 [113]. When Bach1 promotes the increase in intracellular unstable iron, Nrf2 can inhibit the increase in intracellular iron and the increase in the intracellular Bach/Nrf2 ratio can cause iron apoptosis [114]. Therefore, EMT can be linked to ferroptosis through Bach1 (Figure 2).

### 4.3. Wnt-Related Signaling Pathway

The Wnt signaling pathway, which is closely related to EMT, is activated when the Wnt ligands Wnt3a and Wnt1 are secreted and bound to the frizzled receptors (FZD) and a low-density lipoprotein (LRP) co-receptor. LRP receptors phosphorylated by CK1 and GSK3β can recruit the disheveled (Dvl) protein to the plasma membrane, polymerize with Dvl and be activated. Dvl polymer destroys and inactivates the complex by sequestration in multibubbles. Dvl-1 can bind directly to axin, inhibit axin-mediated GSK3β-dependent phosphorylation of β-catenin and lead to the stabilization and accumulation of β-catenin which is then translocated into the nucleus [115,116,117]. In addition, E-cadherin also inhibits β-catenin nuclear translocation by forming the E-cadherin/β-catenin complex. β-catenin and T-cell factor/lymphoid enhancer factor (TCF/LEF) transcription factors act as transcription coactivators that can induce pro-fibrosis gene expression [118]. E-cadherin is downregulated as a marker of the epithelial phenotype, and α-SMA and type I collagen are upregulated as a marker of myofibroblasts in pulmonary fibrosis [119].

The Wnt signaling pathway plays an important role in EMT induction and the development of pulmonary fibrosis [120]. A typical Wnt pathway can be activated by irradiation, but in the presence of DKK-1, the nuclear localization of active β-Catenin decreases. DKK-1 inhibits EMT in vitro and in vivo through Wnt/β-Catenin signaling, inhibits the upregulation of vimentin expression induced by ionizing radiation in alveolar epithelial cells and then resists pulmonary fibrosis [121]. Synergism to induce EMT between TGF-β1 and the Wnt signaling pathway was observed. The expression of an ECM metalloproteinase inducer induced by TGF-β1 can stimulate the production of matrix metalloproteinases (MMPs) in interstitial fibroblasts via the Wnt/β-Cantenin signaling pathway in AEC2 cells [8]. MMPs can degrade all components of ECM, including zinc-dependent endopeptidase; they are expressed at low levels in normal adult tissues but are significantly increased in patients with pulmonary fibrosis [122]. This suggests that MMPs play an important role in the development of pulmonary fibrosis. MMP3 can mediate fibrosis and inhibit distal epithelial repair by activating the β-Catenin and TGF-β pathways [123].

The Wnt signaling pathway also plays a role in ferroptosis. Iron overload produces ROS and LPO, which leads to ferroptosis, thus weakening classical Wnt signaling and inhibiting osteoblast differentiation. Desferramine can reduce iron deposition in cells, reduce ROS and alleviate osteoblast differentiation [124]. Activation of Wnt signaling can reduce the production of lipid peroxidation and ROS in gastric cancer cells, thereby inhibiting ferroptosis in gastric cancer cells. The β-catenin /TCF4 transcription complex directly binds to the promoter region of GPX4 and induces its expression, thereby inhibiting ferroptosis [125].

Some studies have shown that ferroptosis is related to the Wnt pathway and EMT is regulated by the Wnt pathway to play an anti-pulmonary fibrosis role. However, there is no experiment to show the relationship between ferroptosis and pulmonary fibrosis in the Wnt signaling pathway. However, in ovarian cancer cells, the overexpression of FZD7 (a transmembrane receptor) activates the carcinogen Tp63 which increases GPX4 and protects the cells induced by chemotherapy and oxidative stress [126]. Alveolar cells could be particularly vulnerable to abnormal Wnt signaling, because diverse differentiation and death-inducing signals, including p53, p21waf1 and transactivating isoforms of p63, are simultaneously expressed in repairing alveoli after injury, thus inducing pulmonary fibrosis [127]. In lung tissue, silicon stimulation can significantly activate Wnt5a and other inflammatory signaling pathways. Furthermore, IL-6, TNF-α, and other inflammatory factors are released; the expression of GPX4 protein is inhibited and ferroptosis is promoted [77]. The release of inflammatory cytokines activates lung fibroblasts and induces the secretion and deposition of ECM components, resulting in an interstitial fibrotic scar that contributes to impaired gas exchange [128].Therefore, it is worth studying whether ferroptosis regulates pulmonary fibrosis through EMT in the Wnt signaling pathway (Figure 3).

## 5. Summary and Outlook

Iron is an essential substance for the human body, but excess iron enhances the Fenton response and depletes the antioxidant capacity of cells, which leads to ROS accumulation and then induces EMT in alveolar epithelial cells. The type-II alveolar epithelial cells are transformed first into fibroblasts and then into myofibroblasts, producing ECM and finally progressing to pulmonary fibrosis. Ferroptosis can be controlled by equilibrium iron steady state and oxidation steady state, or GPX4 can be affected. At the same time, the purpose of regulating EMT against pulmonary fibrosis has been achieved. Nrf2, TGF-β/Smad, Wnt and other pathways are related to GPX4 and ROS. Importantly, ferroptosis regulation of EMT plays an important role in the prevention and treatment of pulmonary fibrosis. The progression of pulmonary fibrosis can be slowed down by regulating EMT through Bach1, ZEB1, NDRG1, erastin, liproxstatin-1, Cav-1 and other ferroptosis-related genes or proteins. Infection with SARS-CoV-2 can also promote ferroptosis, leading to acute lung inflammation and pulmonary fibrosis. Therefore, the regulation of ferroptosis may provide a novel approach for the prevention and treatment of pulmonary fibrosis (Figure 4).

## Figures and Tables

**Figure 1 biomedicines-11-00163-f001:**
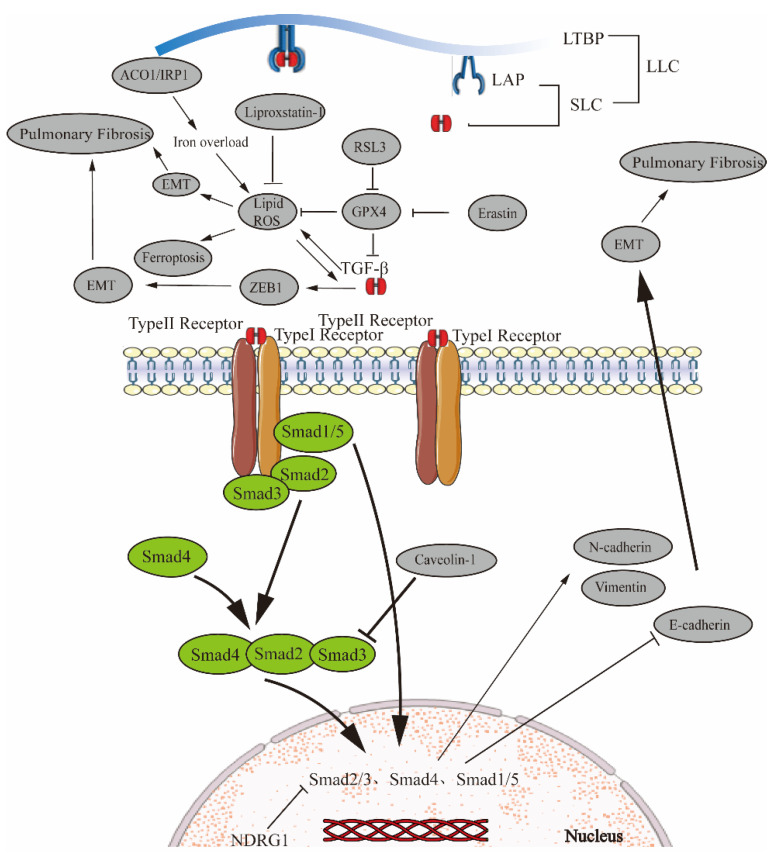
Ferroptosis regulates EMT in pulmonary fibrosis via the TGF-β/Smad signaling pathway. TGF-β non-covalently binds to the latency-associated peptide (LAP) to form a small latent complex (SLC) that blocks TGF-β from binding to its receptor. The SLC connects with LTBP in the endoplasmic reticulum to form the large latent complex (LLC). After TGF-β release by LAP, TGF-β binds to TGFβR-II to phosphorylate TGFβR-II, activates TGFβR-I and phosphorylates intracellular Smad2 and Smad3. Phosphorylated Smad2 and Smad3 interact with Smad4 and translocate to the nucleus, and regulate target gene transcription by binding to DNA-binding transcription factors. The ferroptosis activator RSL3 can inhibit GPX4, thereby increasing ROS, promoting ferroptosis and idiopathic pulmonary fibrosis. Ferroptosis inhibitors liproxstatin-1 inhibit ROS and have the opposite effect. Meanwhile, TGF-β enhances ZEB1-mediated EMT to promote pulmonary fibrosis. ACO1/IRP1 plays a role in regulating intracellular iron homeostasis. Cav-1 attenuates Smad3 nuclear translocation, and the NDRG1 gene inhibits pulmonary fibrosis by inhibiting the Smad2/3 phosphorylation gene. The Smads gene promotes the expression of N-cadherin and Vimentin and inhibits E-cadherin, thus promoting EMT.

**Figure 2 biomedicines-11-00163-f002:**
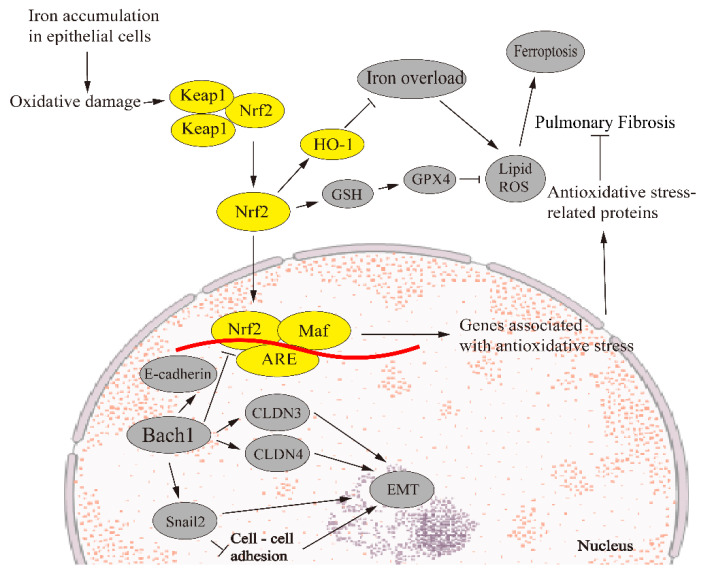
Ferroptosis regulates EMT in pulmonary fibrosis via the Nrf2 signaling pathway. Under normal conditions, Keap1 binds to Nrf2, which is degraded by ubiquitination. However, when iron accumulates in epithelial cells, the imbalance between oxidation and antioxidant reaction results in oxidative stress. Nrf2 dissociates from Keap1 during oxidative stress, and Nrf2 enters the nucleus to form heterodimerization with small Maf or Jun proteins. This dimer binds to ARE, transcriptionally activating its target genes and restoring cell homeostasis. Bach1 regulates EMT through CLDN3 and CLDN4, and Bach1 regulates FOXA1 and snail2, forming GRN to regulate EMT. Bach1 decreased the expression of the E-cadherin gene and promoted the occurrence of EMT. Meanwhile, Bach1 directly activates snail2, which promotes EMT by inhibiting cell–cell adhesion. Knockdown of Nrf2 significantly reduces heme oxygenase 1 (HO-1), promotes the accumulation of lipid peroxidation and causes ferroptosis.

**Figure 3 biomedicines-11-00163-f003:**
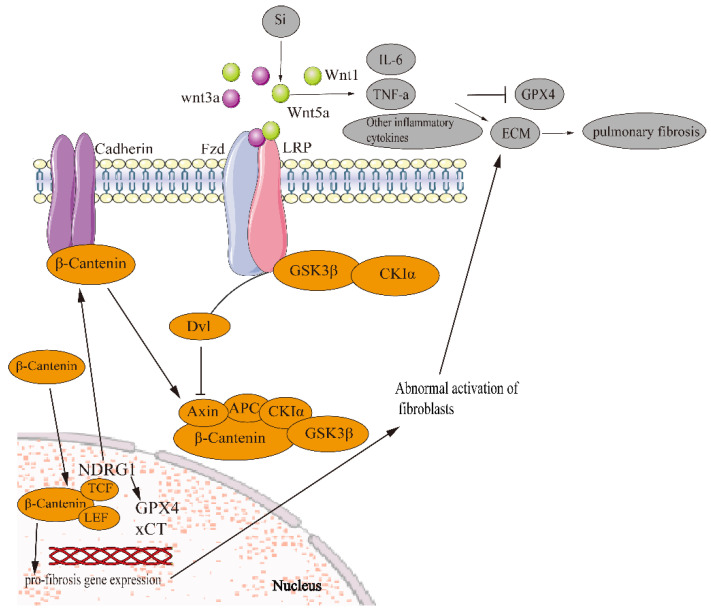
Ferroptosis regulates EMT in pulmonary fibrosis via the Wnt signaling pathway. β-catenin forms a destructive complex with Axin, APC, CKIα and GSK3β. β-catenin is phosphorylated by GSK3β and CKIα, resulting in β-Catenin degradation by ubiquitination. Undegraded β-catenin accumulates in the cytoplasm and is transferred to the nucleus. β-catenin binds to TCF/LEF, which promotes transcription of downstream pro-fibrosis genes, leading to abnormal activation of fibroblasts and ECM deposition. Silicon stimulation can significantly activate Wnt5a and other inflammatory signaling pathways, release IL-6, TNF-α and other inflammatory factors, inhibit the expression of GPX4 protein and promote ferroptosis. When Wnt ligands are secreted and bind to the FZD receptor and LRP coreceptor, the LRP receptor is phosphorylated by CK1α and GSK3β. The phosphorylated LRP receptor recruits Dvl protein into the plasma membrane, and Dvl aggregates and is activated. A Dvl polymer destroys and inactivates the complex. Dvl-1 inhibits axin-mediated GSK3β-dependent phosphorylation of β-catenin, leading to stabilization and accumulation of β-catenin, which is then translocated into the nucleus. E-cadherin also inhibits β-catenin by forming the E-cadherin/β-catenin complex to prevent nuclear translocation of β-catenin. The NDRG1 gene promotes the formation of the E-cadherin/β-catenin adhesion complex on the cell membrane to prevent β-catenin nuclear translocation.

**Figure 4 biomedicines-11-00163-f004:**
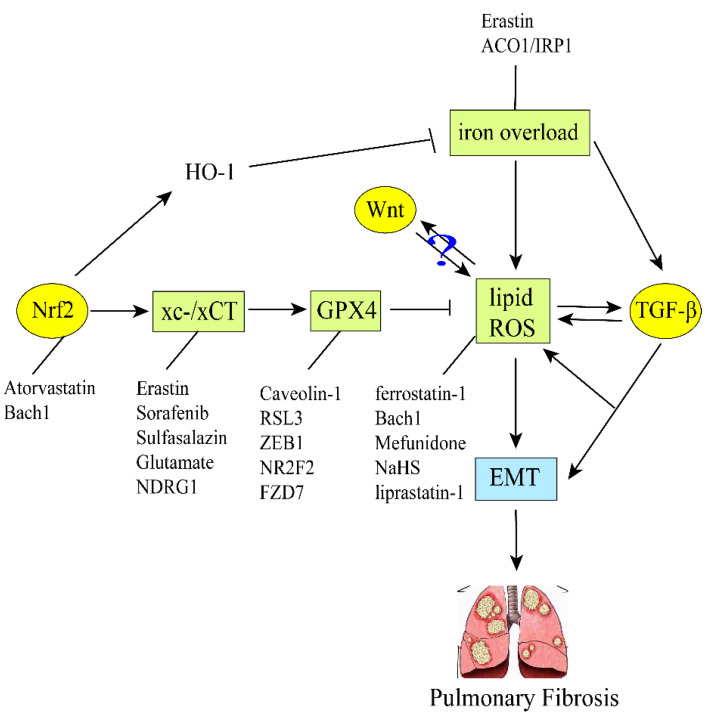
Ferroptosis regulates EMT-induced pulmonary fibrosis in the TGF-β/Smad, Nrf2 and Wnt signaling pathways. GPX4 and system xc- are ferroptosis markers. The decrease of GPX4 increases lipid and ROS. Various regulatory factors regulate system xc-, GPX4, ROS, respectively, and then induce EMT, finally leading to pulmonary fibrosis. Among them, the TGF-β and Nrf2 signaling pathways are associated with it to some extent, while the Wnt signaling pathway remains to be explored.

## Data Availability

Not applicable.

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
