# Peer review of "Role of Ferroptosis in Regulating the Epithelial–Mesenchymal Transition in Pulmonary Fibrosis"

_biomedicines, 2023, doi:10.3390/biomedicines11010163_

Round 1
Reviewer 1 Report
Comments for authors:
The authors for this paper write a review of the current literature on how the process of ferroptosis may be involved in the regulation of EMT in pulmonary fibrosis.
General comments:
The topic that the authors have chosen to review is an important and interesting one, however a lot of work is required to make it readable and suitable for the journal.
Specific comments/questions:
· Authors need to confirm that they have the right name of the process they are reviewing here. Should EMT be epithelial-mesenchymal ‘transition’ or ‘transformation’? They use ‘transition’ in the abstract and then use ‘transformation’ in the remainder of the manuscript.
· I’m not sure ferroptosis can still be defined as new, researchers have been publishing on it for 10 years now.
· There are a lot of grammatical errors in the manuscript, as well as repetition which makes the work difficult to read e.g lines 59-71 has grammatical errors, words missing and also needs more details (such as how do the ferroptosis markers listed have a strong regulatory effect on EMT?).
· Authors write about ROS production and that this leads to ferroptosis several times in the manuscript, however ROS production can lead to several different cell death pathways, and authors don’t make enough of a distinction between how or why ROS specifically leads to ferroptosis and not other cell death pathways in pulmonary fibrosis.
· Several paragraphs have statements/several sentences that don’t fit what the paragraph is describing or are not relevant to the topic being talked about.
o E.g. “iron cells that transport oxygen become prone to death or die” line 113 doesn’t follow on from the previous statements in the paragraph.
· Figures: need some work to show what the authors are describing (and legends describing what the figure shows with the abbreviations explained).
o Figure 1 is titled “Ferroptosis regulates EMT in pulmonary fibrosis via the TGF-b/Smad signalling pathway” however, the figure doesn’t show this. The figure shows what role iron may play in the signalling pathway, but not ferroptosis. The pathway can not be called ferroptosis if you don’t show a link to how this induces lipid peroxidation and subsequent cell death.
§ Liproxstatin-1 (spelled incorrectly in the text)
§ Figure should also show how the expression of Smad leads to EMT.
o Figure 2 – also has some missing elements:
§ Again the same point about the figure not showing ferroptosis regulating EMT, but how the separate elements of iron overload and oxidative damage may interact. How does lipid peroxidation fit into this?
§ Where is the oxidative damage coming from?
§ The red dots in the figure make the text unclear (‘nucleus’ is not very clear).
Author Response
请参阅附件。
Please see the attachment.

Reviewer 2 Report
Ling et al., summarized the important role of ferroptosis in regulating EMT in PF. The authors did a good job of grouping and summarizing previous literature. I enjoyed reading through their review; here are some suggestions to improve the quality of their manuscript and attract a wider category of readers:
Major:
1- The role of ferroptosis in acute lung injury should be discussed.
2- The authors should discuss the role of ferroptosis in post-COVID-19 pulmonary fibrosis as it will be interesting to most of the readers.
3- If there any previous trials for inhibiting ferroptosis and its effect on the development of pulmonary fibrosis?, if yes, those trials should be discussed in this review.
Minor:
1- ]] mark at the beginning of both lines 83 and 157 should be deleted.
